# Capturing Graphs with Hypo-Elliptic Diffusions

**Csaba Toth**[*]
Mathematical Institute
University of Oxford
toth@maths.ox.ac.uk

**Darrick Lee**[*]
Mathematical Institute
University of Oxford
leed@maths.ox.ac.uk

**Celia Hacker**
Department of Mathematics
EPFL
celia.hacker@epfl.ch

**Harald Oberhauser**
Mathematical Institute
University of Oxford
oberhauser@maths.ox.ac.uk

## Abstract

Convolutional layers within graph neural networks operate by aggregating information about local neighbourhood structures; one common way to encode such substructures is through random walks. The distribution of these random walks evolves according to a diffusion equation defined using the graph Laplacian. We extend this approach by leveraging classic mathematical results about hypo-elliptic diffusions. This results in a novel tensor-valued graph operator, which we call the hypo-elliptic graph Laplacian. We provide theoretical guarantees and efficient low-rank approximation algorithms. In particular, this gives a structured approach to capture long-range dependencies on graphs that is robust to pooling. Besides the attractive theoretical properties, our experiments show that this method competes with graph transformers on datasets requiring long-range reasoning but scales only linearly in the number of edges as opposed to quadratically in nodes.

## 1   Introduction

Obtaining a latent description of the non-Euclidean structure of a graph is central to many applications. One common approach is to construct a set of features for each node that represents the local neighborhood of this node; pooling these node features then provides a latent description of the whole graph. A classic way to arrive at such node features is by random walks: at the given node one starts a random walk, and extracts a summary of the local neighbourhood from its sample trajectories. We revisit this random walk construction and are inspired by two classical mathematical results:

**Hypo-elliptic Laplacian.** In the Euclidean case of Brownian motion $B = (B_t)_{t \geq 0}$ evolving in $\mathbb{R}^n$, the quantity $u(t, x) = \mathbb{E}[f(B_t)|B_0 = x]$ solves the heat equation $\partial_t u = \Delta u$ on $[0, \infty) \times \mathbb{R}^n$, $u(0, x) = f(x)$. Seminal work of Gaveau [23] in the 1970s shows that if one replaces $f(B_t)$ in the expectation by a functional of the whole trajectory, $F(B_s : s \in [0, t])$, then a path-dependent heat equation can be derived where the classical Laplacian $\Delta$ is replaced by the hypo-elliptic Laplacian.

**Free Algebras.** A simple way to capture a sequence – for us, a sequence of nodes visited by a random walk on a graph – is to associate with each sequence element an element in an algebra[1] and multiply these algebra elements together. If the algebra multiplication is commutative,

---

[*]Equal contribution; order determined by random coin flip.

[1]An algebra is a vector space where one can multiply elements; e.g. the set of $n \times n$ matrices with matrix multiplication. This multiplication can be non-commutative; e.g. $A \cdot B \neq B \cdot A$ for general matrices $A, B$.

36th Conference on Neural Information Processing Systems (NeurIPS 2022).

the sequential structure is lost but if it is non-commutative, this captures the order in the sequence. In fact, by using the free associative algebra, this can be done faithfully and linear functionals of this algebra correspond to functionals on the space of sequences.

We leverage these ideas from the Euclidean case of $\mathbb{R}^d$ to the non-Euclidean case of graphs. In particular, we construct features for a given node by sampling from a random walk started at this node, but instead of averaging over the end points, we average over path-dependent functions. Put informally, instead of asking a random walker that started at a node, *"What do you see now?"* after $k$ steps, we ask *"What have you seen along your way?"*. The above notions from mathematics about the hypo-elliptic Laplacian and the free algebra allow us to formalize this in the form of a generalized graph diffusion equation and we develop algorithms that make this a scalable method.

**Related Work.** From the ML literature, [49, 27] popularized the combination of deep learning architectures to capture random walk histories. Such ideas have been incorporated, sometimes implicitly, into graph neural networks (GNN) [53, 9, 54, 18, 29, 5, 33] that in turn build on convolutional approaches [37, 38, 27], as well as their combination with attention or message passing [45, 65, 24], and more recent improvements [72, 46, 43, 15] that provide and improve on theoretical guarantees. Another classic approach are graph kernels, see [7] for a recent survey; in particular, the seminal paper [36] explored the connection between diffusion equations and random walkers in a kernel learning context. More recently, [14] proposed sequence kernels to capture the random walk history. Furthermore, [17] uses the signature kernel maximum mean discrepancy (MMD) [16] as a metric for trees which implicitly relies on the algebra of tensors that we use, and [48] aggregates random walk histories to derive a kernel for graphs. Moreover, the concept of network motifs [44, 55] relates to similar ideas that describe a graph by node sequences. Further, the Bethe Hessian [52] has been successfully used in spectral clustering and shares the same goal of capturing pathdependence via a "deformed Laplacians", although the mathematical approach is very different to ours. Directly related to our approach is the topic of learning diffusion models [35, 13, 60, 20, 11] on graphs. While similar ideas on random walks and diffusion for graph learning have been developed by different communities, our proposed method leverages these perspectives by capturing random walk histories through a novel diffusion operation.

Our main mathematical influence is the seminal work of Gaveau [23] from the 1970s that shows how Brownian motion can be lifted into a Markov process evolving in a free algebra to capture path-dependence. This leads to a heat equation governed by the hypo-elliptic Laplacian. These insights had a large influence in PDE theory, see [51, 31], but it seems that their discrete counterpart on graphs has received no attention despite the well-developed literature on random walks on graphs and general non-Euclidean objects, [68, 19, 26, 63]. A key challenge to go from theory to application is handling the computational complexity. To do so, we build on ideas from [62] to design effective algorithms for the hypo-elliptic graph diffusion.

**Contribution and Outline.** We introduce the hypo-elliptic graph Laplacian which allows to effectively capture random walk histories through a generalized diffusion model.

- In Section 3, we introduce the hypo-elliptic variants of standard graph matrices such as the adjacency matrix and (normalized) graph Laplacians. These hypo-elliptic variants are formulated in terms of tensor-valued matrices rather than scalar-valued matrices, and can be manipulated using linear algebra in the same manner as the classical setting.

- The hypo-elliptic Laplacian leads to a corresponding diffusion model, and in Theorem 1, we show that the solution to this generalized diffusion equation summarizes the microscopic picture of *the entire history of random walks* and not just their location after $k$ steps.

- This solution provides a rich description of the local neighbourhood about a node, which can either be used directly as node features or be pooled over the graph to obtain a latent description of the graph. Theorem 2 shows that these node features characterize random walks on the graph, and we provide an analogous statement for graph features in Appendix E.

- One can solve the hypo-elliptic graph diffusion equation directly with linear algebra, but this is computationally prohibitive and Theorem 3 provides an efficient low-rank approximation.

- Finally, Section 5 provides experiments and benchmarks. A particular focus is to test the ability of our model to capture long-range interactions between nodes and the robustness of pooling operations which makes it less susceptible to the "over-squashing" phenomenon [2].

## 2 Sequence Features by Non-Commutative Multiplication.

We define the *space of sequences in $\mathbb{R}^d$* by

$$\mathsf{Seq}(\mathbb{R}^d) := \bigcup_{k=0}^{\infty} (\mathbb{R}^d)^{k+1},$$

where elements are sequences denoted by $\mathbf{x} = (x_0, x_1, \ldots, x_k) \in (\mathbb{R}^d)^{k+1}$. Assume we are given an injective map, which we call the *algebra lifting*,

$$\varphi : \mathbb{R}^d \to H.$$

from $\mathbb{R}^d$ into an algebra $H$. We can use this to define a *sequence feature map*[2]

$$\widetilde{\varphi} : \mathsf{Seq}(\mathbb{R}^d) \to H, \quad \widetilde{\varphi}(\mathbf{x}) = \varphi(\delta_0 \mathbf{x}) \cdots \varphi(\delta_k \mathbf{x}), \tag{1}$$

where $\delta_0 \mathbf{x} = x_0$ and $\delta_i \mathbf{x} := x_i - x_{i-1}$ for $i \geq 1$ are used to denote the *increments* of a sequence $\mathbf{x} = (x_0, \ldots, x_k)$. This map associates to any sequence $\mathbf{x} \in \mathsf{Seq}(\mathbb{R}^d)$ an element of the algebra $H$. If the multiplication in $H$ is commutative, then the map $\widetilde{\varphi}$ would have no information about the order of increments, i.e. $\varphi(\delta_0 \mathbf{x}) \cdots \varphi(\delta_k \mathbf{x}) = \varphi(\delta_{\pi(0)} \mathbf{x}) \cdots \varphi(\delta_{\pi(k)} \mathbf{x})$ for any permutation $\pi$ of $\{0, \ldots, k\}$. However, if the multiplication in $H$ is "non-commutative enough" we expect $\widetilde{\varphi}$ to be injective.

**A Free Construction.** Many choices for $H$ are possible, but intuitively it makes sense to use the "most general object" for $H$. The mathematically rigorous approach is to use the *free algebra over $\mathbb{R}^d$* and we give a summary in Appendix B. Despite this abstract motivation, the algebra $H$ has a concrete form: it is realized as a sequence of tensors in $\mathbb{R}^d$ of increasing degree, and is defined by

$$H := \{\mathbf{v} = (\mathbf{v}_0, \mathbf{v}_1, \mathbf{v}_2, \ldots) : \mathbf{v}_m \in (\mathbb{R}^d)^{\otimes m}, \ m \in \mathbb{N}, \ \|\mathbf{v}\| < \infty\},$$

where by convention $(\mathbb{R}^d)^{\otimes 0} = \mathbb{R}$, and we describe the norm $\|\mathbf{v}\|$ in the paragraph below. For example, if $\mathbf{v} = (\mathbf{v}_m)_{m \geq 0} \in H$, then $\mathbf{v}_0$ is a scalar, $\mathbf{v}_1$ is a vector, $\mathbf{v}_2 \in (\mathbb{R}^d)^{\otimes 2}$ is a $d \times d$ matrix, and so on. The vector space structure of $H$ is given by addition and scalar multiplication according to

$$\mathbf{v} + \mathbf{w} := (\mathbf{v}_m + \mathbf{w}_m)_{m \geq 0} \in H \quad \text{and} \quad \lambda \mathbf{v} := (\lambda \mathbf{v}_m)_{m \geq 0} \in H$$

for $\lambda \in \mathbb{R}$, and the algebra structure is given by

$$\mathbf{v} \cdot \mathbf{w} := \left( \sum_{i=0}^{m} \mathbf{v}_i \otimes \mathbf{w}_{m-i} \right)_{m \geq 0} \in H. \tag{2}$$

**An Inner Product.** If $e^1, \ldots, e^d$ is a basis of $\mathbb{R}^d$, then every tensor $\mathbf{v}_m \in (\mathbb{R}^d)^{\otimes m}$ can be written as

$$\mathbf{v}_m = \sum_{1 \leq i_1, \ldots, i_m \leq d} c_{i_1, \ldots, i_m} e^{i_1} \otimes \cdots \otimes e^{i_m}.$$

This allows us to define an inner product $\langle \cdot, \cdot \rangle_m$ on $(\mathbb{R}^d)^{\otimes m}$ by extending

$$\langle e^{i_1} \otimes \cdots \otimes e^{i_m}, e^{j_1} \otimes \cdots \otimes e^{j_m} \rangle_m = \begin{cases} 1 & : i_1 = j_1, \ldots, i_m = j_m, \\ 0 & : \text{otherwise.} \end{cases} \tag{3}$$

to $(\mathbb{R}^d)^{\otimes m}$ by linearity. This gives us an inner product on $H$,

$$\langle \mathbf{v}, \mathbf{w} \rangle := \sum_{m \geq 0} \langle \mathbf{v}_m, \mathbf{w}_m \rangle_m$$

such that $H$ is a Hilbert space; in particular we get a norm $\|\mathbf{v}\| := \sqrt{\langle \mathbf{v}, \mathbf{v} \rangle}$. To sum up, the space $H$ has a rich structure: it has a vector space structure, it has an algebra structure (a noncommutative product), and it is a Hilbert space (an inner product between elements of $H$ gives a scalar).

---

[2]There are variants of this sequence feature map, which are discussed in Appendix G.

**Characterizing Random Walks.**    From Equation (1), we have constructed a map $\widetilde{\varphi}$ that maps a sequence $\mathbf{x} \in \mathsf{Seq}(\mathbb{R}^d)$ of arbitrary length into the space $H$ (see Appendix C for further details). Our aim is to apply this to the sequence of node attributes corresponding to random walks on a graph. Therefore, the expectation of $\widetilde{\varphi}$ should be able to characterize the distribution of the random walk. Formally the map $\widetilde{\varphi}$ is *characteristic* if the map $\mu \mapsto \mathbb{E}_{\mathbf{x} \sim \mu}[\widetilde{\varphi}(\mathbf{x})]$ from the space of probability measures on $\mathsf{Seq}(\mathbb{R}^d)$ into $H$ is injective. Indeed, if the chosen lifting $\varphi$ satisfies some mild conditions this holds for $\widetilde{\varphi}$; see Appendix C and [16, 62].

**Linear Functionals.**    The quantity $\mathbb{E}_{\mathbf{x} \sim \mu}[\widetilde{\varphi}(\mathbf{x})]$ characterizes the probability measure $\mu$ but is valued in the infinite-dimensional Hilbert space $H$. Using the inner product, we can instead consider

$$\langle \boldsymbol{\ell}, \mathbb{E}_{\mathbf{x} \sim \mu}[\widetilde{\varphi}(\mathbf{x})] \rangle \text{ for } \boldsymbol{\ell} = (\boldsymbol{\ell}_0, \boldsymbol{\ell}_1, \boldsymbol{\ell}_2, \ldots, \boldsymbol{\ell}_M, 0, \ldots) \in H \text{ and } M \geq 1 \tag{4}$$

which is equivalent to knowing $\mathbb{E}_{\mathbf{x} \sim \mu}[\widetilde{\varphi}(\mathbf{x})]$; i.e. the set (4) characterizes $\mu$. This is analogous to how one can use either the moment generating function of a real-valued random variable or its sequence of moments to characterize its distribution; the former is one infinite-dimensional object (a function), the latter is a infinite sequence of scalars. We extend a key insight from [62] in Section 4: a linear functional $\langle \boldsymbol{\ell}, \mathbb{E}_{\mathbf{x} \sim \mu}[\widetilde{\varphi}(\mathbf{x})] \rangle$ can be efficiently approximated without directly computing $\mathbb{E}_{\mathbf{x} \sim \mu}[\widetilde{\varphi}(\mathbf{x})]$ or storing large tensors.

**The Tensor Exponential.**    While we will continue to keep $\varphi$ arbitrary for our main results (see [62] and Appendix G for other choices), we will use the *tensor exponential* $\exp_{\otimes} : \mathbb{R}^d \to H$, defined by

$$\exp_{\otimes}(x) = \left( \frac{x^{\otimes m}}{m!} \right)_{m \geq 0}, \tag{5}$$

as the primary example throughout this paper and in the experiments in Section 5. With this choice, the induced sequence feature map is the discretized version of a classical object in analysis, called the path signature, see Appendix C.

## 3   Hypo-Elliptic Diffusions

Throughout this section, we fix a labelled graph $\mathcal{G} = (\mathcal{V}, \mathcal{E}, f)$, that is $\mathcal{V}$ is a set of $n$ nodes $\mathcal{V} = \{1, \ldots, n\}$, $\mathcal{E}$ denotes edges and $f : \mathcal{V} \to \mathbb{R}^d$ is the set of continuous node attributes[3] which map each node to an element in the vector space $\mathbb{R}^d$. Two nodes $i, j \in \mathcal{V}$ are *adjacent* if $(i, j) \in \mathcal{E}$ is an edge, and we denote this by $i \sim j$. The *adjacency matrix* $A$ of a graph is defined by $A_{i,j} = 1$, whenever $i \sim j$, and $0$ otherwise. We denote by $\deg(i)$ the number nodes that are adjacent to node $i$.

**Random Walks on Graphs.**    Let $(B_k)_{k \geq 0}$ be the simple random walk on the nodes $\mathcal{V}$ of $\mathcal{G}$, where the initial node is chosen uniformly at random. The *transition matrix* of this time-homogeneous Markov chain is

$$P_{i,j} := \mathbb{P}(B_k = j | B_{k-1} = i) = \begin{cases} \frac{1}{\deg(i)} & : i \sim j \\ 0 & : \text{otherwise.} \end{cases}$$

Denote by $(L_k)_{k \geq 0}$ the random walk lifted to the node attributes in $\mathbb{R}^d$, that is

$$L_k := f(B_k). \tag{6}$$

Recall that the *normalized graph Laplacian* for random walks is defined as $\mathcal{L} = I - D^{-1}A$, where $D$ is diagonal degree matrix; in particular, the entry-wise definition is

$$\mathcal{L}_{i,j} := \begin{cases} -\frac{1}{\deg(i)} & : i \sim j \\ 1 & : i = j \\ 0 & : \text{otherwise.} \end{cases}$$

The discrete graph diffusion equation for $U_k \in \mathbb{R}^{n \times d}$ is given by

$$U_k - U_{k-1} = -\mathcal{L}U_{k-1}, \quad U_0^{(i)} = f(i) \tag{7}$$

---

[3]The labels given by the labelled graph are called *attributes*, while the computed updates are called *features*.

where the initial condition $U_0 \in \mathbb{R}^{n \times d}$ is specified by the node attributes.[4] The probabilistic interpretation of the solution to this diffusion equation is classical and given as

$$U_k = (\mathbb{E}[L_k \mid B_0 = i])_{i=1}^n = P^k U_0. \tag{8}$$

This allows us to compute the solution $u_k$ using the transition matrix $P = I - \mathcal{L}$.

**Random Walks on Algebras.** We now incorporate the history of a random walker by considering the quantity

$$\mathbb{E}[\widetilde{\varphi}(\mathbf{L}_k) \mid B_0 = i] = \mathbb{E}[\varphi(\delta_0 \mathbf{L}) \cdots \varphi(\delta_k \mathbf{L}) \mid B_0 = i] \tag{9}$$

where $\mathbf{L}_k = (L_0, \dots, L_k)$. Since $\widetilde{\varphi}$ captures the whole history of the random walk $\mathbf{L}_k$ over node attributes, we expect this expectation to provide a much richer summary of the neighborhood of node $i$ than $\mathbb{E}[L_k|B_0 = i]$. The price is however, the computational complexity, since (9) is $H$-valued. We first show, that analogous to (7), the quantity (9) satisfies a diffusion equation that can be computed with linear algebra. To do so, we develop a graph analogue of the hypo-elliptic Laplacian and replace the scalar entries of the matrices with entries from the algebra $H$.

**Matrix Rings over Algebras.** We first revisit the adjacency matrix $A \in \mathbb{R}^{n \times n}$ and replace it by the *tensor adjacency matrix* $\widetilde{A} = (\widetilde{A})_{i,j} \in H^{n \times n}$, that is $\widetilde{A}$ is a matrix but instead of scalar entries its entries are elements in the algebra $H$. The matrix $A$ has an entry at $i, j$ if nodes $i$ and $j$ are connected; $\widetilde{A}$ replaces the $i, j$ entry with an element of $H$ that tells us how the node attributes of $i$ and $j$ differ,

$$\widetilde{A}_{i,j} := \begin{cases} \varphi(f(j) - f(i)) & : i \sim j \\ 0 & : \text{otherwise.} \end{cases} \tag{10}$$

Matrix multiplication works for elements of $H^{n \times n}$ by replacing scalar multiplication by multiplication in $H$, that is $(\widetilde{B} \cdot \widetilde{C})_{i,j} = \sum_{k=1}^n \widetilde{B}_{i,k} \cdot \widetilde{C}_{k,j}$ for $\widetilde{B}, \widetilde{C} \in H^{n \times n}$ and $\widetilde{B}_{i,k} \cdot \widetilde{C}_{k,j}$ denotes multiplication in $H$ as in Equation (2). For the classical adjacency matrix $A$, the $k$-th power counts the number of length $k$ walks in the graph, so that $(A^k)_{i,j}$ is the number of walks of length $k$ from node $i$ to node $j$. We can take powers of $\widetilde{A}$ in the same way as in the classical case, where

$$(\widetilde{A}^k)_{i,j} = \sum_{\mathbf{x}} \varphi(\delta_1 \mathbf{x}) \cdots \varphi(\delta_k \mathbf{x})$$

where the sum is taken over all length $k$ walks $\mathbf{x} = (f(i), \dots f(j))$ from node $i$ to node $j$ (full details are provided in Appendix D). Since $\widetilde{\varphi}(\mathbf{x})$ characterizes each walk $\mathbf{x}$, the entry $\widetilde{A}_{i,j}^k$ can be interpreted as a summary of all walks which connect nodes $i$ and $j$.

**Hypo-elliptic Graph Diffusion.** Similar to the tensor adjacency matrix, we define the *hypo-elliptic graph Laplacian* as the $n \times n$ matrix

$$\widetilde{\mathcal{L}} = I - D^{-1} \widetilde{A} \in H^{n \times n},$$

where $D$ is the degree matrix embedded into $H^{n \times n}$ at tensor degree 0. The entry-wise definition is

$$\widetilde{\mathcal{L}}_{i,j} := \begin{cases} \frac{-\varphi(f(j) - f(i))}{\deg(i)} & : i \sim j \\ 1 & : i = j \\ 0 & : \text{otherwise.} \end{cases} \tag{11}$$

We can now formulate the *hypo-elliptic graph diffusion equation* for $\mathbf{v}_k \in H^n$ as

$$\mathbf{v}_k - \mathbf{v}_{k-1} = -\widetilde{\mathcal{L}} \mathbf{v}_{k-1}, \quad \mathbf{v}_0^{(i)} = \varphi(f(i)). \tag{12}$$

Analogous to the classic graph diffusion (8), the hypo-elliptic graph diffusion (12) has a probabilistic interpretation in terms of $\mathbf{L}$ as shown in Theorem 1 (the proof is given in Appendix D).

**Theorem 1.** *Let $k \in \mathbb{N}$, $\mathbf{L}_k = (L_0, \dots, L_k)$ be the lifted random walk from (6), and $\widetilde{P} = I - \widetilde{\mathcal{L}}$ be the tensor adjacency matrix. The solution to the hypo-elliptic graph diffusion equation (12) is*

$$\mathbf{v}_k = (\mathbb{E}[\varphi(\delta_1 \mathbf{L}_k) \cdots \varphi(\delta_k \mathbf{L}_k)|B_0 = i])_{i=1}^n = \widetilde{P}^k \mathbb{1}_H.$$

*Furthermore, if $F \in H^{n \times n}$ is the diagonal matrix with $F_{i,i} = \varphi(f(i))$, then*

$$F \mathbf{v}_k = (\mathbb{E}[\widetilde{\varphi}(\mathbf{L}_k)|B_0 = i])_{i=1}^n.$$

---

[4]The attributes over all nodes are given by an $n \times d$ matrix; in particular $U_k^{(i)}$ is the $i^{\text{th}}$ row of the matrix.

In the classical diffusion equation, $U_k$ captures the concentration of the random walkers after $k$ time steps over the nodes. In the hypo-elliptic diffusion equation, $\mathbf{v}_k$ captures summaries of random walk histories after $k$ time steps over the nodes since $\widetilde{\varphi}(\mathbf{L}_k)$ summarizes the whole trajectory $\mathbf{L}_k = (L_0, \ldots, L_k)$ and not only the endpoint $L_k$.

**Node Features and Graph Features.** Theorem 1 can then be used to compute features $\Phi(i) \in H$ for individual nodes as well as a feature $\Psi(\mathcal{G})$ for the entire graph. The former is given by $i$-th component $\mathbf{v}_k^{(i)}$ of the solution $\mathbf{v}_k = (\mathbf{v}_k^{(i)})_{i=1,\ldots,n} \in H^n$ of Equation (12),

$$\Phi(i) := \mathbf{v}_k^{(i)} = \mathbb{E}[\widetilde{\varphi}(\mathbf{L}_k) \,|\, B_0 = i] = (F\widetilde{P}^k \mathbf{v}_0)^{(i)} \in H,$$

since the random walk $B$ chooses the starting node $B_0 = i$ uniformly at random. The latter can be computed by mean pooling the node features, which also has a probabilistic interpretation as

$$\Psi(\mathcal{G}) := \frac{1}{n} \sum_{i=1}^{n} \mathbf{v}_k^{(i)} = \mathbb{E}[\widetilde{\varphi}(\mathbf{L}_k)] = n^{-1}(\mathbb{1}_H^T F \widetilde{P}^k \mathbf{v}_0) \in H, \tag{13}$$

where $\mathbb{1}_H^T := (1_H, \ldots, 1_H) \in H^n$ is the all-ones vector in $H$ and $1_H$ denotes the unit in $H$.

**Characterizing Graphs with Random Walks.** The graph and node features obtained through the hypo-elliptic diffusion equation are highly descriptive: they characterize the entire history of the random walk process if one also includes the time parametrization, as described in Appendix C.

**Theorem 2.** *Suppose $\Psi$ is the graph feature map from Equation (13) induced by the tensor exponential algebra lifting including time parametrization. Let $\mathcal{G}$ and $\mathcal{G}'$ be two labelled graphs, and $\mathbf{L}_k = (L_0, \ldots, L_k)$ and $\mathbf{L}'_k = (L'_0, \ldots, L'_k)$ be the $k$-step lifted random walk as defined in Equation (6). Then, $\Psi(\mathcal{G}) = \Psi(\mathcal{G}')$ if and only if the distributions of $\mathbf{L}_k$ and $\mathbf{L}'_k$ are equal.*

It is instructive to contrast this result with the classical diffusion case; the latter only uses the marginal distribution of $L_k$ to capture the graph structure, which at least intuitively has much less expressive power. Indeed, in Appendix E, we show that for elementary graphs, this already leads to big differences in expressive power. Further, an analogous result holds for the node features, and we prove both results in Appendix E. While we use the tensor exponential in this article, many other choices of $\widetilde{\varphi}$ are possible and result in graph and node features with such properties: under mild conditions, if the algebra lifting $\varphi : \mathbb{R}^d \to H$ characterizes measures on $\mathbb{R}^d$, the resulting node feature map $\Phi$ characterizes the random walk, see [62], which in turn implies the above results. Possible variations are discussed in Appendix G.

**General (Hypo-elliptic) Diffusions and Attention.** One can consider more general diffusion operators, such as the normalized Laplacian $\mathcal{K}$ of a weighted graph. We define its lifted operator $\widetilde{\mathcal{K}} \in H^{n \times n}$ analogous to Equation (11), resulting in a generalization of Theorem 1 with $\widetilde{\mathcal{K}}$ replacing $\widetilde{\mathcal{L}}$. In the flavour of convolutional GNNs [8], we consider a weighted adjacency matrix $A \in \mathbb{R}^{n \times n}$

$$A_{i,j} = \begin{cases} c_{i,j} & : i \sim j \\ 0 & : \text{otherwise,} \end{cases}$$

for $c_{i,j} > 0$. The corresponding normalized Laplacian $\mathcal{K}$ is given by $\mathcal{K} = I - D^{-1}A$, where $D$ is a diagonal matrix with $D_{i,i} = \sum_{j \in \mathcal{N}(i)} c_{i,j}$. A common way to learn the coefficients is by introducing parameter sharing across graphs by modelling them as $c_{i,j} = \exp(a(f(i), f(j)))$ using a local attention mechanism, $a : \mathbb{R}^d \times \mathbb{R}^d \to \mathbb{R}$ [65]. In our implementation, we use additive attention [4] given by $a(f(i), f(j)) = \mathsf{LeakyRelu}_{0.2}(W_s f(i) + W_t f(j))$, where $W_s, W_t \in \mathbb{R}^{1 \times d}$ are linear transformations for the source and target nodes, but different attention mechanisms can also be used; e.g. scaled dot-product attention [64]. Then, the corresponding transition matrix $P = D^{-1}A$ is defined as $P_{ij} = \mathrm{softmax}_{k \in \mathcal{N}(i)}(a(f(i), f(k)))_j$. The lifted transition matrix is defined as

$$\widetilde{P} = \begin{cases} P_{i,j}\varphi(f(j) - f(i)) & : i \sim j \\ 0 & : \text{otherwise.} \end{cases}$$

The statements of Theorem 1 immediately generalize to this variation by replacing the expectation with respect to a non-uniform random walk. Hence, in this case the use of attention can be interpreted as learning the transition probabilities of a random walk on the graph.

# 4 Efficient Algorithms for Deep Learning

The previous sections show that the node feature $\Phi(i)$ provides a structured description of the neighborhood of node $i$ and it is instructive to think of a linear functional $\langle \ell, \Phi(i) \rangle$ as answering a specific question about the node neighbourhood, see Appendix E for examples. The naive computation of $\langle \ell, \Phi(i) \rangle$ by first computing $\Phi(i)$ and taking the inner product is too expensive, especially when $\ell = (\ell_0, \ldots, \ell_M, 0, \ldots) \in H$ for large $M$. To address this we revisit two observations from [62]: first, for a rank-1 functional $\ell \in H$, the computation of $\langle \ell, \Phi(i) \rangle$ is computationally cheap. Second, restriction to small $M$ limits the expressive power but can be counteracted by composition: any choice of $d$ different functionals $\ell^1, \ldots, \ell^d \in H$ gives a label update $f(i) \mapsto (\langle \ell^j, \Phi(i) \rangle)_{j=1,\ldots,d} \in \mathbb{R}^d$ for the graph. Repeating such a label update a few times with low-degree $M$ and rank-1 functionals turns out to be as powerful as computing one update for general functionals with arbitrary high $M$. The first observation should not be too surprising given the popularity of low rank approximations; the second observation is reminiscent to constructing a high-degree polynomial by composing low-degree polynomials[5] or the width-vs-depth phenomenon in neural nets and we give more details below.

**Computing Rank-1 Functionals.**    First, we focus on a *rank*-1 linear functional $\ell \in H$ given as

$$\ell = (\ell_m)_{m \geq 0} \text{ with } \ell_m = u_{M-m+1} \otimes \cdots \otimes u_M \text{ and } \ell_m = 0 \text{ for } m > M, \qquad (14)$$

where $u_m \in \mathbb{R}^d$ for $m = 1, \ldots, M$ for a fixed $M \geq 1$. Theorem 3 shows that for such $\ell$, the computation of $\langle \ell, \hat{\Phi}(i) \rangle$, where $\hat{\Phi}(i)$ is the node feature without the basepoint, can be done (a) efficiently by factoring this low-rank structure into the recursive computation, and (b) simultaneously for all nodes $i \in \mathcal{V}$ in parallel. This can then be used to compute rank-$R$ functionals for $R > 1$, and for $\langle \ell, \Phi(i) \rangle$; see Appendix F, where we also provide a pseudocode implementation.

**Theorem 3.** *Let $\ell$ be as in* (14) *and define $f_{k,m} \in \mathbb{R}^n$ for $m = 1, \ldots, M$ as*

$$f_{1,m} := \frac{1}{m!} \left( P \odot C^{u_{M-m+1}} \odot \cdots \odot C^{u_M} \right) \cdot \mathbb{1},$$

*where $\mathbb{1}^T := (1, \ldots, 1) \in \mathbb{R}^n$ is the all-ones vector; and for $2 \leq k$ and $1 \leq m \leq M$ recursively as*

$$f_{k,m} := P \cdot f_{k-1,m} + \sum_{r=1}^m \frac{1}{r!} \left( P \odot C^{u_{M-m+1}} \odot \cdots \odot C^{u_{M-m+r}} \right) \cdot f_{k-1,m-r}, \qquad (15)$$

*where the matrix $C^u = (C^u_{i,j}) \in \mathbb{R}^{n \times n}$ is defined as*

$$C^u_{i,j} := \begin{cases} \langle u, f(j) - f(i) \rangle & : i \sim j, \\ 0 & : otherwise. \end{cases}$$

*Here $\odot$ denotes element-wise[6] multiplication, while $\cdot$ denotes matrix multiplication. Then, it holds for $i \in \mathcal{V}$, random walk length $k \in \mathbb{Z}_+$, and tensor degree $m = 1, \ldots, M$, that*

$$f_{k,m}(i) = \langle \ell_m, \hat{\Phi}_k(i) \rangle,$$

*where $\hat{\Phi}_k(i) = \mathbb{E}[\varphi(\delta_1 \mathbf{L}_k) \cdots \varphi(\delta_k \mathbf{L}_k) \,|\, B_0 = i]$.*

Overall, Eq. (15) computes $f_{k,m}(i)$ for all $i \in \mathcal{V}$, $k = 1, \ldots, K$, $m = 1, \ldots, M$ in $O(K \cdot M^2 \cdot N_E + M \cdot N_E \cdot d)$ operations, where $N_E \in \mathbb{N}$ denotes the number of edges; see App. F. In particular, one does not need to compute $\Phi(i) \in H$ directly or store large tensors.

**Graph Labelling Layers.**    Fixing $d$ rank 1-functionals $\ell^1, \ldots, \ell^d \in H$ induces a label update $f(i) \mapsto (\langle \ell^i, \Phi(i) \rangle)_{i=1,\ldots,d} \in \mathbb{R}^d$. Theorem 3 allows us to compute this update in parallel for all nodes in $\mathcal{V}$. Such a label update is similar to hidden layer in a NN and we can stack such updates, see Figure 3 in App. H.1. As in NN, the $d$ functionals in each "graph labelling layer" are optimized by gradient descent. Finally, note that a rank $R$ functional is the sum of $R$ rank-1 functionals so we can immediately carry out the same construction with rank-$R$ functionals by adding a mixing layer.

---

[5]For example, $1 + x + x^2$ composed with $1 + 2x^2$ yields the degree 4 polynomial $1 + (1 + 2x^2) + (1 + 2x^2)^2$.

[6]For example $\begin{bmatrix} 1 & 2 \\ 3 & 4 \end{bmatrix} \odot \begin{bmatrix} 5 & 6 \\ 7 & 8 \end{bmatrix} = \begin{bmatrix} 5 & 12 \\ 21 & 32 \end{bmatrix}$.

To sum up, a graph labelling layer is determined by the random walk length $k$, the maximal tensor degree $M$, maximal tensor rank $R$ and the functionals are then found by optimization.

Using a single layer of low-rank functionals limits the expressiveness but stacking layers allows in practice to approximate general, high-degree $M$ functionals. Some theoretical results can be found in [62]; however, here we simply appeal to the analogy with NN where stacking simple transformations provides a flexible functional class with good inductive bias.

## 5 Experiments

We implemented the above approach and call the resulting model **G**raph**2T**ens **N**etworks since it represents the neighbourhood of a node as a sequence of tensors, which is further pushed through a low-tensor-rank constrained linear mapping, similarly to how neural networks linearly transform their inputs pre-activation. A conceptual difference is that in our case the non-linearity is applied first and the projection secondly, albeit the computation is coupled between these steps. We provide further experiments and ablation studies of our models in Appendix H.

**Experimental Setup.** The aim of our main experiment is to test the following key properties of our model: (1) ability to capture long-range interactions between nodes in a graph, (2) robustness to pooling operations, hence making it less susceptible to the "over-squashing" phenomenon [2]. We do this by following the experiments in [70]. In particular, we show that our model is competitive with previous approaches for retaining long-range context in graph-level learning tasks but without computing all pairwise interactions between nodes, thus keeping the influence distribution localized [73]. We further give a detailed ablation study to show the robustness of our model to various architectural choices in Appendix H.2. As a second experiment, we follow the previous applications of diffusion approaches to graphs that have mostly considered inductive learning tasks, e.g. on the citation datasets [13, 60, 11]. Our experimentation on these datasets are available in Appendix H.3, where the model performs on par with short-range GNN models, but does not seem to benefit from added long-range information a-priori. However, when labels are dropped in a $k$-hop sanitized way as in [50], the performance decrease is less pronounced.

**Datasets.** We use two biological graph classification datasets (NCI1 and NCI109), that contain around $\sim$4000 biochemical compounds represented as graphs with $\sim$30 nodes on average [67, 1]. The task is to predict whether a compound contains anti-lung-cancer activity. The dataset is split in a ratio of $80\% - 10\% - 10\%$ for training, validation and testing. Previous work [2] has found that GNNs that only summarize local structural information can be greatly outperformed by models that are able to account for global contextual relationships through the use of *fully-adjacent* layers. This was further improved on by [70], where a local neighbourhood encoder consisting of a GNN stack was upgraded with a Transformer submodule [64] for learning global interactions.

**Model Details.** We build a GNN architecture primarily motivated by the GraphTrans (small) model from [70], and only fine-tune the pre- and postprocessing layers(s), random walk length, functional degree and optimization settings. In detail, a preprocessing MLP layer with 128 hidden units is followed by a stack of 4 `G2TN` layers each with RW length-5, max rank-128, max tensor degree-2, all equipped with JK-connections [73] and a max aggregator. Afterwards, the node features are combined into a graph-level representation using gated attention pooling [41]. The pooled features are transformed using a final MLP layer with 256 hidden units, and then fed into a softmax classification layer. The pre- and postprocessing MLP layers employ skip-connections [30]. Both MLP and `G2TN` layers are followed by layer normalization [3], where GTN layers normalize their rank-1 functionals independently across different tensor degrees, which corresponds to a particular realization of group normalization [69]. We randomly drop $10\%$ of the features for all hidden layers during training [58]. The attentional variant, `G2T(A)N` also randomly drops $10\%$ of its edges and uses 8 attention heads [65]. Training is performed by minimizing the categorical cross-entropy loss with an $\ell_2$ regularization penalty of $10^{-4}$. For optimization, Adam [32] is used with a batch size of 128 and an inital learning rate of $10^{-3}$ that is decayed via a cosine annealing schedule [42] over 200 epochs. Further intuition about the model and architectural choices are available in Appendix H.1.

**Baselines.** We compare against (1) the baseline models reported in [70], (2) variations of Graph-Trans, (3) other recently proposed hierarchical approaches for long-range graph tasks [50]. Groups

Table 1: Comparison of classification accuracies on NCI biological datasets, where we report mean and standard deviation over 10 random seeds for our models.

| Model | GNN Type | GNN Count | NCI1 (%) | NCI109 (%) |
|---|---|---|---|---|
| Set2Set [40, 66] | GCN | 3 | $68.6 \pm 1.9$ | $69.8 \pm 1.2$ |
| SortPool [40, 75] | GCN | 3 | $73.8 \pm 1.0$ | $74.0 \pm 1.2$ |
| SAGPool$_h$ [40] | GCN | 3 | $67.5 \pm 1.1$ | $67.9 \pm 1.4$ |
| SAGPool$_g$ [40] | GCN | 3 | $74.2 \pm 1.2$ | $74.1 \pm 0.8$ |
| GIN [21, 72] | GIN | 8 | $80.0 \pm 1.4$ | - |
| GCN + VN [74, 24] | GCN | 2 | 71.5 | - |
| HGNet-EdgePool [74, 54] | GCN+RGCN | $3 + 2$ | 77.1 | - |
| HGNet-Louvain [74, 54] | GCN+RGCN | $3 + 2$ | 75.1 | - |
| GIN + FA [2, 72] | GIN | 8 | $81.5 \pm 1.2$ | - |
| GraphTrans (small) [70, 64] | GCN | 3 | $81.3 \pm 1.9$ | $79.2 \pm 2.2$ |
| GraphTrans (large) [70, 64] | GCN | 4 | $82.6 \pm 1.2$ | $82.3 \pm 2.6$ |
| **G2T(A)N** (ours) | G2T(A)N | 4 | $\mathbf{81.9 \pm 1.2}$ | $78.0 \pm 2.3$ |
| **G2TN** (ours) | G2TN | 4 | $80.7 \pm 2.5$ | $\mathbf{78.9 \pm 2.5}$ |

of models in Table 1 are separated by dashed lines if they were reported in separate papers, and the first citation after the name is where the result first appeared. The number of GNN layers in HGNet are not discussed by [50], and we report it as implied by their code. We organize the models into three groups divided by solid lines: (a) baselines that only apply neighbourhood aggregations, and hierarchical or global pooling schemes, (b) baselines that first employ a local neighbourhood encoder, and afterwards fully densify the graph in one way or another so that all nodes interact with each other *directly*, (c) our models that we emphasize thematically belong to (a).

**Results.** In Table 1, we report the mean and standard deviation of classification accuracy computed over 10 different seeds. Overall, both our models improve over all baselines in group (a) on both datasets, maximally by $1.9\%$ on NCI1 and by $4.8\%$ on NCI109. In group (b), G2T(A)N is solely outperformed by GraphTrans (large) on NCI1 by only $0.7\%$. Interestingly, the attention-free variation, G2TN , performs better on NCI109, where it performs very slightly worse than GraphTrans (small).

**Discussion.** The previous experiments demonstrate that our approach performs very favourably on long-range reasoning tasks compared to GNN-based alternatives without global pairwise node interactions. Several of the works we compare against have focused on extending GNNs to larger neighbourhoods by specifically designed graph coarsening and pooling operations, and we emphasize two important points: (1) our approach can efficiently capture large neighbourhoods without any need for coarsening, (2) it already performs well with simple mean-pooling as justified by Theorem 2 and experimentally supported by the ablation studies in Appendix H.2. Although the Transformer-based GraphTrans slightly outperforms our model potentially due to its ability to learn global interactions, it is not entirely clear how much of the global graph structure it is able to infer from interactions of short-range neighbourhood summaries. Finally, Transformer models can be bottlenecked by their quadratic complexity in nodes, while our approach only scales with edges, and hence, it can be more favourable for large sparse graphs in terms of computations.

## 6 Conclusion

Inspired by classical results from analysis [23], we introduce the hypo-elliptic graph Laplacian. This yields a diffusion equation and also generalizes its classical probabilistic interpretation via random walks but now taking history into account. In addition to several attractive theoretical guarantees, we provide scalable algorithms. Our experiments show that this can lead to largely improved baselines for long-range reasoning tasks. A promising future research theme is to develop improvements for the classical Laplacian in this hypo-elliptic context; including lazy random walks [71]; nonlinear diffusions [12]; and source/sink terms [60]. Another theme could be to extend the geometric study [61] of over-squashing to this hypo-elliptic point of view which is naturally tied to sub-Riemannian geometry [59]. A limitation in our theoretical results is that for the iterations of low-rank approximations only partial results exist and expanding this is an interesting (algebra-heavy) topic.

## Acknowledgments and Disclosure of Funding

Csaba Toth was supported by a Mathematical Institute Award from the University of Oxford. Celia Hacker and Darrick Lee were supported by NCCR-Synapsy Phase-3 SNSF grant number 51NF40-185897. Darrick Lee and Harald Oberhauser were supported by the Hong Kong Innovation and Technology Commission (InnoHK Project CIMDA). Harald Oberhauser was also supported by the DataSig Program [EP/S026347/1] and the Oxford-Man Institute.

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
