# OpenReview forum: "Capturing Graphs with Hypo-Elliptic Diffusions"
_NeurIPS.cc/2022/Conference — NeurIPS 2022 Accept_

### Official Review · Reviewer_zTgE · 2022-06-25

**Rating:** 6
**Confidence:** 4
**Soundness:** 4 excellent
**Presentation:** 3 good
**Contribution:** 3 good

**Summary:**

The paper introduces hypo-elliptic graph Laplacians -- a generalization of classical graph Laplacians to higher order tensors -- that are capable of storing the entire history of a given random-walk. This is studied in the context of attributed graphs where each node is equipped with vector information. Theoretical analysis is carried over to demonstrate that 1) the analogous tensor-like diffusion process retains the probabilistic interpretation of standard scalar random walks and 2) graph feature maps can be defined on these processes w/o losing expressive power. A low-rank approximation is provided to make the approach scalable and experiments are conducted on datasets that present long-range dependencies.

**Questions:**

- While the problem is relevant, the solution is somewhat not surprising. If we store more information directly via tensors, then this will avoid aggregating information from distant nodes along the 1-hops only hence alleviating the over-squashing phenomenon. A perhaps more interesting analysis should entail what this higher-order approach is effectively accomplishing. Is it `just' a matter of storing higher-order (i.e. non-local) information? Do we actually need random walks exactly or a higher-order approach that store information without leveraging the algebra representation is going to accomplish similar results? What does such method give us when compared to more standard MPNNs like, say GIN, when augmented with JK connections for example? I don't expect a specific answer here but some speculation might improve the message.
- In the proof of Theorem 3 I believe there is a typo in the proof of the case for $f_{1,m}$ where we should have $\frac{1}{d_{i}m!}\prod_{r= M - m +1}^{M}\langle u_{r},f(j)-f(i)\rangle$.
- Some analysis of the number of parameters might be relevant here. How do you compare to baselines?
-  A clock experiment comparing running time of GT2N/GT2(A)N with baselines would also be important here.
- The size of the random walk is chosen to be 5 on the molecular datasets: how would that compare to the average diameter of the graphs? Is in fact this quite a `global' approach already?
- Why in the experiment section you only compare with [64] on some of the datasets used in [64] and not all of them? A comparison with Table 2 in [64] would appear meaningful here.


Conclusions: The paper is novel and the idea of relying on the algebra representations to memorize the full history of the attributed random walk is an elegant formalization of an idea that has entered existing `tricks' one way or another. Of relevance the proposed approximation in Theorem 3. There are a few questions partly left-unanswered. As mentioned above, how the questions are addressed may affect the final score (in both ways).

**Limitations:**

Societal impact is not reported. Limitations are partly addressed in the paper.

**Strengths And Weaknesses:**

Contribution and novelty: To my knowledge -- I am not an expert in the field of hypo-elliptic operators on non-Euclidean domains -- the contribution of the paper is original. This applies to both the theoretical analysis in Theorem 1,2 and the low-rank approximation in Theorem 3 that builds on previous work in [56].

Presentation: The paper is very well-presented and the organisation does not require main modifications. On a minor side, some of the paper is quite technical and potentially obscure to a significant part of the community.

Strengths (in random order):

- The exposition is clear.
- The problem at hand is of interest, given that standard MPNNs are known to struggle on tasks with long-range dependencies, mainly due to the problem of over-squashing.
- The theoretical analysis in Theorem 3 is of relevance in the context of the tools and techniques introduced, given that it provides a scalable approach.
- The implementation/experiment section has detailed explanation with sufficient ablation studies that strengthen the message and indeed highlight the robustness of the method.


Weaknesses (in random order) -- I reserve some unanswered points to the Questions paragraph below; depending on the rebuttal such points could be solved or become weaknesses with consequent adjusting of the score.

- Theorem 3 inevitably leads to a loss in expressive power and it is not entirely clear how the loss would depend on M, and R (minor point).
- The proposed method is not (significantly) better than faster/standard GNN methods on task like node-classification where long-range dependencies may not be crucial. This is somewhat expected but it limits the impact of the approach.

---

> ### Author Response · Authors · 2022-08-02
> **Author Response**
>
> Thank you very much for the careful reading and questions!
>
> 1. Theorem 3 (..) loss in expressive power" That is true, using rank $R$ functionals with small degree $M$ instead of general functional significantly reduces the expressive power; however, iterating such simple functionals counteracts this to some extent (at least intuitively and backed partially by some theoretical results). We have rewritten Section 4 to emphasize this more.
>
> 2. "not (significantly) better than faster/standard GNN methods on task like node-classification where long-range dependencies may not be crucial." Yes, we agree, the strength of the current approach is a principled approach for situations where long-range dependencies matter, while keeping the model to be local. We try to emphasize this more in the updated version. We also emphasize that the complexity of our model is $O(E)$ rather than $O(N^2)$ in GraphTrans. We empirically demonstrate this in the new Table 6 in Appendix H.4.
>
> 3. "Do we actually need random walks exactly or a higher-order approach (...) accomplish similar results" This is a great question that we do not have an exact answer to. While it is certainly true that there exist other "higher-order" methods that should perform well in theory, this is often not the case in practice, for example k-WL GNNs are theoretically more powerful, but often underperform in practice compared to MPNNs, and their computational cost is also more overbearing. Other related approaches are network motifs and graph substructure counts, that can provide improvements for MPNNs, see e.g. (Bouritsas et al 2022), but in the general case substructure enumeration is computationally prohibitive. We believe our main contribution is a mathematically principled framework for propagating higher-order information within a neighbourhood through tensorized summary of random walks with a computational cost that is at least comparable to classical MPNNs.
>
> Bouritsas, Giorgos, et al. "Improving graph neural network expressivity via subgraph isomorphism counting." IEEE Transactions on Pattern Analysis and Machine Intelligence (2022).
>
> 4. "compared to more standard MPNNs like, say GIN..." This is another great question, which the reviewer has partially answered themselves. It is indeed true that JK connections somewhat help in the training of deeper GNNs by aiding in the propagation of the gradient into the lower layers. However, these approaches proceed by only aggregating information within the 1-hop neighbourhoods, followed by pointwise activation functions. After several iterations of this, the information that reaches a node from its more distant neighbours is oversquashed due to compressing a combinatorially growing amount of information into a fixed size vector representation. In contrast, we summarize information directly within the $k$-hop neighbourhood by explicitly encoding both short and long-range interactions using tensors, making the approach less biased towards only retaining short-range information. We would like to draw a parallel with recurrent neural networks for sequential data. Classical formulations of RNNs often fail to retain long-range information, but it has been demonstrated that tensor-based representations perform better at such tasks by explicitly representing the space of interactions globally. In some sense, one could think of the tensor product (or the tensor convolution product in our case) as a better nonlinearity for encoding long-range information as compared to the standard pointwise activations used in RNNs/MPNNs. Although not a complete answer, we hope this somewhat addresses the reviewer's question, and we are happy to discuss this further.
>
> 5. "typo in the proof" Thanks for looking at the proofs and for catching the typo, you are entirely correct.
>
> 6. "Some analysis of the number of parameters" For the NCI datasets, the G2TN and G2T(A)N models have 505k and 519k parameters respectively. The only benchmark which reported number of parameters is GraphTrans (small), which has 0.5M parameters, and GraphTrans (large), which has 4.2M parameters. When compared to GraphTrans (small), our models achieves similar performance with a comparable number of parameters, while our complexity is $O(E)$ rather than $O(N^2)$.
>
> 7. "clock experiment comparing running time" Thank you for this suggestion! We have provided computation times for various graphs and model parameters in Table 6 and 7 in Appendix H.4.
>
> continued..

---

> > ### Author Response · Authors · 2022-08-02
> > **Author Response cont.**
> >
> > 8. "size of the random walk is chosen to be 5". The average diameters of the graphs in the dataset are reported in the new Table 8 in Appendix H.6. For the NCI datasets, the average diameter of graphs is roughly 13. We have experimented with various walk lengths up to $k=10$, and found that $k=5$ performed the best with the current architecture. We call our approach "local" in the sense that each update layer aggregates local neighbourhood information as opposed to transformer models or virtual node approaches in which all nodes are able to directly communicate.
> >
> > 9. "compare with [64] on other datasets" As the reviewer requested, we have updated our implementation and started experiments on the OGB-Molpcba benchmark. We have run the exact same model as on the NCI datasets with a lower learning rate of $10^{-4}$. We get a validation-test AP of $0.269$-$0.252$. While this is not at the level of GraphTrans at the moment, the model they use has around 4.2M parameters, while the model we have used has only 597k. Due to the size of the dataset and the time constraints, we do not find it feasible to experiment with different ways to upscale our model, but these results still demonstrate that the proposed long-range interaction encoding layers can provide out-of-the-box improvements over classical GNNs without virtual nodes, such as GCN and GIN.
> >
> > 10. "Limitations" We added a short paragraph on broader impacts at the beginning of the appendix. Please let us know if you believe a more detailed discussion would be helpful. Another limitation is the fact that theoretical properies about stacking low rank functionals of such sequence features is a difficult problem, and is an interesting direction for future research. We have added a sentence at the end of the conclusion regarding this point.

---

> ### Comment · Reviewer_zTgE · 2022-08-04
> **Response to rebuttal**
>
> Thank you for clarifications, answers and additional experiments.
>
> I think the paper still has a few limitations, mainly concerning experimentation (node-classification task, comparison with big graph-Transformers) and the theoretical analysis leaves quite a few questions open, regarding expressive power when using low-rank approximations (and perhaps in the broader landscape of higher-order models).
>
> The framework itself is still a `higher-order’ one and in some way goes beyond the boundaries imposed by an MPNN; this still leaves open the important question of whether MPNNs really cannot deal with long-range dependencies. Also, I am always a bit unsure whenever I see improvements that are ascribed to the underlying model being able to capture long-range dependencies. This is a subtle but important distinction that is not entirely clear: to what extent we are making distant nodes interact with each other vs storing information and then “counting”?
>
> Despite that, I still think this paper has its own valuable contributions. I have decided to increase my score to 6 and I am recommending acceptance.
>
> One final question: concerning the running time experiment, do you also have numbers for the same running time of Graph-Trans (small)?

---

> > ### Author Response · Authors · 2022-08-08
> > **Author Response 2**
> >
> > Thank you very much for taking our response into consideration! Just a couple of comments on your new response:
> >
> > 1. "limitations concerning theoretical analysis" We agree that there are still open questions about the theoretical analysis, which we think would be very interesting for future work. However, we wanted to point out there are some partial theoretical results about stacking such low-rank functionals in the Appendix of [62]. In particular, Proposition C.3 in [62] shows that iterated low degree tensors can represent higher degree tensors in the first layer; thus recovering some of the expressivity lost due to the low-rank projection. Due to the heavier algebraic background necessary (half/quasi-shuffle algebras) to properly discuss such results, we opted to refer the reader to the literature instead. We hope that this current work will lead to more research into this interesting area.
> >
> > 2. "to what extent we are making distant nodes interact with each other vs storing information and then “counting”?" This is a great question. The more direct interpretation is the latter: within each layer, we are essentially storing information about distributions of random walks, and then "counting" sequential events by applying low rank functionals. One could argue that distant nodes "interact" with each other via the learning procedure for the low rank functionals (ie. *which* sequential events are counted).
> >
> > 3. "concerning the running time experiment, do you also have numbers for the same running time of Graph-Trans (small)?" Unfortunately, due to time constraints, we weren't able to perform the runtime experiments for GraphTrans.

---

### Official Review · Reviewer_f1qr · 2022-07-11

**Rating:** 6
**Confidence:** 2
**Soundness:** 3 good
**Presentation:** 4 excellent
**Contribution:** 3 good

**Summary:**

In this work the authors present a formalism where a tensor valued version of the graph laplacian, which they call the hypo-elliptic graph laplacian, is used to model the distribution of node attribute propagation via random walks as a diffusion process. They show how the solutions for the state of the resulting system after length k walks can be computed via low rank approximation algorithms and compare a concrete realization of their method to other common architectures for graph modeling. They argue that their method excels in representing long range dependencies while scaling more favorably than other approaches in the size of the graph.

**Questions:**

1. Is there a reason why $m$ and $k$ are not used a bit more interchangeably? It seems like that dimension of cross product space and the sequence lengths we are lifting are practically the same?
2. Can you describe the meaningful difference if any of thinking of elements of H as the m-dimensional cross of the node attributes along the path, rather than each sequence element individually as lifted to $(\mathbb{R}^{d})^{\otimes m}\$? (I think no, this is part of the elegance of the setup?)

**Limitations:**

1. Somewhat duplicated from above, I am concerned about the $m=2$ restriction of the implementation results presented. It seems like this may take the power out of the method in some sense by barely maintaining the added depth/tensorial nature beyond the classical diffusion equation using the normal graph laplacian. Further, and this is _just a conjecture_, but it is possible that this limitation on $m$, plus the rank-1 approximation scheme is responsible for the shakeout of the results wrt the GraphTrans competitive baseline. Interested to hear the authors discuss this further.

2. The scalability of the approach may be problematic given that the complexity analysis already relies on the Rank-1 approximation and also contains an $M^2$ term suggesting that the small M setting may be a requirement for practical implementations. Without demonstration on graphs of realistic scales (well beyond 30 nodes), as well as the resultant larger graph diameters and path lengths, it's unclear whether the approach has to always be nearly reduced to the classical setting in order to applied to graphs of non-trivial sizes.

**Strengths And Weaknesses:**

###  Strengths

**Quality/Clarity**: Sections 1-3 (along with the relevant appendices) are very well written, and do a remarkable job at providing an understanding of the background machinery required to understand the proposed method. In particular, the punchline results through Eq. 11 and 12 about the theoretical expressivity in node and graph representation extractable from the solution to the hypo-elliptic graph diffusion equation are exciting.

**Originality**: Benefit of the doubt is granted on the overall presentation of the formalism being novel (reviewer is not a theorist), though I assume novelty is mostly concentrated on the tensor valued version of the diffusion equation, Eq 11.

### Weaknesses

**Clarity**: The work is remarkably readable, but there are a couple points:
- At the moment one reads line 105, it _is not_ immediately clear that we have built the map between $Seq(\mathbb{R}^d)$ and $H$, rather it should cite Appendix lines 612-618 as these are sort of the final step of that construction.
- The section on the low rank approximation of the functionals and the "Building Neural Networks" section is much harder to understand than the sections that precede it.

**Significance**: Despite the theoretical expressivity of the proposed model class, the particular setting evaluated only explores a max tensor degree of 2. This seems like a severely restricted realization of the model based on my understanding that $m$ sort of represents the dimension of the "summary capturing" capacity of the lifted form of the diffusion solution. The method only performs _competitively_ with GraphTrans. Ideally, since it's being proposed as an alternative to the family of available GNNs, the implementation results will be more developed in future work to more convincingly demonstrate the benefits in practice that the framework argues that it offers in theory.

---

> ### Author Response · Authors · 2022-08-02
> **Author Response**
>
> Thanks for the encouraging words about clarity of presentation, this was our major concern. The reviewer is correct in that our main theoretical contribution is the tensor-valued graph Laplacian along with scalable approximation algorithms adapted to learning from graphs.
>
> 1. "line 105", as suggested, we've added a reference to the equation where $\tilde \varphi$ is introduced; thank you for bringing this to our attention.
>
> 2. "Section on the low-rank approximation" We have completely rewritten the start of this section to emphasize the main ideas. Please let us now if this clarifies it.
>
>
> 3. "why $m$ and $k$ are not used a bit more interchangeably" and "meaningful difference from thinking of elements of $H$ as the m-dimensional cross of the node attributes along the path"
>
> You raise a very important point here, thank you for giving us the opportunity to clarify this. Rather than simply taking the tensor product of node features visited by the random walk, we first lift the node features into the free algebra (using the algebra lifting $\varphi$, which we take to be the tensor exponential throughout), and multiply together the lifted node features using the "tensor convolution product" defined in eq. (2). The outcome of this is a graded description using an infinite sequence of tensors of increasing degrees, that we can truncate at any given degree. In contrast, simply tensoring the node features along a walk of length $k$ would result in exactly one tensor of degree $k$. The former has many theoretical and practical benefits. On the theoretical side, the choice of the tensor exponential directly leads to Theorem 2. One of the practical benefits is that we decompose the space of interactions into various orders, and the truncation degree $M$ need not be the same as the sequence length $k$. This decouples these two parameters, and we have refined control over the complexity of the hypothesis space regardless of the sequence length. For further mathematical benefits, we refer the reviewer to Appendix B in [62].
>
>
> 5. "the restriction to $m=2$" You are completely right that just using $M=2$ functionals would significantly reduce the expressiveness of the resulting graph features. However, we stack/compose such functionals by iterating our layer several times. That is, given a labelled graph we choose $R$ many functionals ($m=2$) to update the labels of each node. Using the results from Section 4 (Theorem 3), this allows us to do such an update very cheaply.
> A loose analogy is with the composition of polynomials: given a degree $2$ polynomial such as $1+x^2$, one can precompose it with another degree-$2$ polynomial, e.g. $2x^2$, which results in the degree-$4$ polynomial $1+4 x^4$. Another analogy is a deep network, where each layer is a fairly simple transformation, but stacking such layers results in very powerful expressive features. There are even some partial theoretical results about stacking low-rank functionals for sequential data in [62].
> We have rewritten the paragraph in Section 4 to emphasize this "stacking" of low-rank layers. We have experimented with values of $M \in [2, 3, 4]$, and the $M=2$ setting is what we found to perform best under the current architecture.
>
> 6. "scalability of the approach" Thank you for giving us the opportunity to clarify this point. We would like to emphasize that choosing larger values of $M$ is not a constraint, and although theoretically the algorithm has a complexity of $M^2$, this is not what is observed in practice. In particular, we bring to the reviewer's attention the newly added Table 7 in Appendix H.4. This shows that in practice, the scaling is even sublinear with respect to $M$. This is because in eq. (15) the computation of $f_{k, m}$ is parallelized over different values $m$ for a fixed $k$. To better demonstrate this, we also added Algorithm 1 in Appendix F to demonstrate the computation flow.

---

> ### Comment · Reviewer_f1qr · 2022-08-09
> **Acknowledgement of the author rebuttal**
>
> Thanks to the authors for the detailed responses and clarification.
>
> In particular, the edits to section 4 are very much appreciated and help illustrate the key correspondence between the formalism and implementation better. The additional table on scaling wrt the expressivity parameters is also appreciated.
>
> I second the other reviewer's comment that this work may be slightly obscure to others in this community but I believe the work is valuable! (due to area of expertise/confidence, I can't comfortably raise my score, but stand by my acceptance recommendation)

---

### Official Review · Reviewer_sq7T · 2022-07-11

**Rating:** 5
**Confidence:** 2
**Soundness:** 3 good
**Presentation:** 3 good
**Contribution:** 3 good

**Summary:**

The authors present a novel version of the graph Laplacian which instead of capture information focused on start/end of a path, it emphasizes information along paths. This is inspired on previous notions of the hypo-elliptic Laplacian. Moreover, since the hypo-elliptic graph laplacian is a tensor-valued graph operator, the authors provide low-rank approximations which allow numerical computations, is it scales linearly with the number of edges, contrary to the standard case which grows quadratically with the number of nodes.

Together with the Hypo-elliptic Laplacian another key ingredient is the perspective of free algebras, which is relevant since the order in which nodes visited in a path is relevant, and free algebras induce algebra multiplication that are non-commutative.



**Questions:**

I understand that the proposed numerical scheme is presented in Theorem 3, but I would suggest to introduce a pseudo code. In general that would allow the readers to have a specific module where one can focus on explicit programatic steps, rather than having to follow non-programmatic terminology that naturally corresponds to a theorem.

Further, the authors describe the complexity to be O(K · M^2 · N_E + M · N_E · d) where $M$ is the maximum tensor degree, $N_E$ is the number of edges, $d$ is the dimension of node attributes. Given all these terms in complexity, would it be possible for the authors to report the execution time? I think this would be relevant for the reader, as for instance, the number of edges in the datasets ($N_E$) is not reported in the paper. Further, it is not that straight forward to track the values for the remaining parameters involved in the algorithm complexity. Providing the parameter values of the algorithm complexity, potentially in the supplementary material,  per dataset and model would be of great benefit.

**Limitations:**

It seems the authors do not provide potential negative societal impacts.

**Strengths And Weaknesses:**

The paper does a great job to briefly introduce the basic notions related to the hypo-elliptic graph and the corresponding free algebra notions that are required to grasp the basic intuitions of the concepts here considered. Further, supporting material is presented in the supplementary material that helps to grasp basic concepts related to this paper.

The presents an interesting modeling approach by showing/motivating that the proposed Laplacian indeed has certain properties that might be appealing in the context of learning long dependencies for GNNs. The analysis of the path, and not only relying on where it starts and where it ends has already been suggested, for instance (in a different context) by the non-back tracking operator (or the corresponding Bethe Hessian), even in the context of spectral clustering, where the graph Laplacian plays a crucial role [A].

Bibliography
[A]. Saade, Alaa, Florent Krzakala, and Lenka Zdeborová. "Spectral clustering of graphs with the bethe hessian." Advances in Neural Information Processing Systems 27 (2014).

---

> ### Author Response · Authors · 2022-08-02
> **Author Response**
>
> Thank you for the encouraging words regarding the presentation. Regarding the points raised
>
> 1. We were not aware of the Bethe Hessian, thank you for bringing it to our attention. Our impression is that this is quite different on a mathematical level and aimed at a different problem, however it shares the same underlying idea to capture path-dependence via a "deformed" Laplacian. We have now updated the related work section to reflect this connection.
>
> 2. Thank you for the suggestion. We emphasize that conceptually, eq. (15) is all one needs to implement our algorithm, since it only uses standard matrix operations, which can be efficiently parallelized over multiple linear functionals, nodes and graphs using a fast vectorized implementation. As suggested, we have added a pseudo-code version of the low-rank algorithm at the end of Appendix F and now make a reference to it before Theorem 3 for those readers who would prefer this formulation. This formulation also allows us to highlight the fact that one of the recursive steps can be parallelized and we empirically demonstrate this parallel speedup in Table 7 in Appendix H.4.
>
> 3. We have added a summary of dataset statistics, along with model parameters used in Appendix H.6. Furthermore, we have provided computation times for various graph and model parameters in Appendix H.4. We hope that these additional details clarify the scalability of our model.
>
> 4. "Limitations": We added a short paragraph on broader impacts at the beginning of the appendix. Please let us know if you believe a more detailed discussion would be helpful.

---

### Official Review · Reviewer_foUs · 2022-07-25

**Rating:** 5
**Confidence:** 4
**Soundness:** 3 good
**Presentation:** 3 good
**Contribution:** 3 good

**Summary:**

In graph neural networks, convolutional layers operate by aggregating information about local neighborhood structures. This paper extends to hypo-elliptic diffusions and the derived tensor valued graph operator.

In short, this paper builds the operators by using random walks but instead of taking only the endpoints of random walks, takes averages over path-dependent functions. Here the ordering of vertices visited in a random walk matters. Using random walks to build feature vectors for graphs is a classical idea (e.g., Deepwalk), which has also been extended to other local graph features such as breadth-first trees etc (e.g., Node2Vec). This paper uses the entire path rather than the endpoint and build an algebraic operator.




**Questions:**

There are a few issues that the authors can possibly address/discuss:

	1. About descriptive power, is there theoretical understanding that the hypo-elliptic diffusion is strictly more powerful than just using regular heat diffusion on graphs?
	2. There seems to be a tradeoff between computational cost vs results of the model via parameters such as k, the length of random walks, and R the rank of the functionals. Could the authors discuss the influence of these parameters in model performance?
	3. The results are comparable to earlier work that uses a Transformer model with graph neural networks (GraphTrans), which captures long-term interactions. The discussion at the end of section 5 regarding the performance comparison of this method and Transformer is a bit wishful. Can the authors provide more solid results backing up the discussion here (e.g., the summaries in this paper can also help Transformer submodule)?


**Strengths And Weaknesses:**

Strength: The approach of extension from using only endpoint of a random walk to the entire path of the random walk is natural. The paper presented a theorem (Theorem 2) which says that two labelled graphs are the same if and only if the distribution of k-step random walks of the two graphs are the same. The authors use diffusion over tensor transition matrix and introduce an attention mechanism. The paper explained how to efficiently calculate the operator on a graph.

Weaknesses: see next box on questions.

---

> ### Author Response · Authors · 2022-08-02
> **Author Response**
>
> Thank you for the careful reading and thoughtful questions. We have addressed them as follows
>
> 1. This is similar to a question by Reviewer zTgE about "expressive power". In short, if one uses a classical simple random walk on labels to diffuse the label information, then the resulting graph features can only rely on the distribution of the random walk after $k$ steps, in our notation $L_k$; in contrast, capturing path dependence (as we do via the algebra) captures the distribution of the whole sequence, see Theorem 2. We have added a sentence after Theorem 2 and a new section at the end of Appendix E that show the difference between using only the marginal distribution vs the whole random walk distribution on elementary examples. Furthermore, note that the information of classical diffusion is contained in the level one component of hypo-elliptic diffusion. Thank you for giving us the opportunity to clarify this point. We believe this is one of the strongest contributions of this paper and we hope this has been clarified.
>
> 2. This is an interesting question, thank you for it. The computational trade-offs are discussed in Section 4, and to demonstrate the practical scaling with respect to important parameters of the layer, we added Tables 6-7 in Appendix H.4. To reiterate the four main hyperparameters of our model are: 1) the length of the random walk $k$, 2) the maximal degree of linear functionals $M$, 3) their rank $R$, and 4) the number of layers. The first three control very different aspects of the functions that are learnable by the model. 1) represents the largest range at which distinct nodes can nonlinearly interact with each other. 2) represents the maximal order of these interactions, similarly to how moments of random variables encode higher order dependencies as one increases their degree. This can be made even more explicit, as the sequence features we use correspond to "ordered moments", see Section 1 of (Kiraly and Oberhauser 2019). 3) for a given maximal degree, the rank controls the complexity of these interactions fixed at a given order. The rank is analogous to the width of neural network layer. Finally, increasing 4), the depth, effectively corresponds to increasing all of these hyperparameters at the same time. In general, increasing any of these should improve the expressivity of the model, although overfitting could become an issue (disregarding the double-descent phenomenon for the moment).
>
> 3. Thank you for this observation. We agree that the sentence in question is unsubstantiated for now and we have removed it from the main text. However, we emphasize that we do everything locally and hence our main contribution is a principled approach to capture the $k$-hop neighbourhood about a node in an expressive and efficient way.
>
> 4. "two labelled graphs are the same if and only if the distribution of k-step random walks of the two graphs are the same" This is not what we claim in Theorem 2. There may be examples of distinct labelled graphs which have the same distribution of k-step random walks. For example, consider the 3-node and 6-node cycle graphs with node labels RGB and RGBRGB respectively; these two graphs will have the same distribution of k-step walks. What we claim in Theorem 2 is that we are able to distinguish between graphs which have different k-step random walks, which we believe is still highly descriptive.
>
> Király, Franz J., and Harald Oberhauser. "Kernels for sequentially ordered data." Journal of Machine Learning Research 20 (2019).

---

### Meta-Review · Area_Chair_pbF6 · 2022-08-27

**Recommendation:** Accept
**Confidence:** Certain

**Metareview:**

This paper proposes a new principled graph learning method to aggregate information local neighborhood. The proposed method essentially aggregates information from random walk but not only w.r.t. end points but also w.r.t. the paths. All reviewers agree that the idea is novel and the idea is well founded. Valid questions regarding expressive power, computational expense, etc. were raised and were addressed reasonably during rebuttal period.

**Award:**

No

---

### Decision · Program_Chairs · 2022-09-14

Accept